# Unraveling the Genetic Heterogeneity of Acute Lymphoblastic Leukemia Based on NGS Applications

**DOI:** 10.3390/cancers16233965

**Published:** 2024-11-26

**Authors:** Valentina Ramírez Maldonado, Josgrey Navas Acosta, Iván Maldonado Marcos, Ángela Villaverde Ramiro, Alberto Hernández-Sánchez, Jesús M. Hernández Rivas, Rocío Benito Sánchez

**Affiliations:** 1Centro de Investigación del Cáncer, IBMCC, CSIC, Universidad de Salamanca, IBSAL (Instituto de Investigación Biomédica de Salamanca) Campus, Miguel de Unamuno, 37007 Salamanca, Spain; vramirem@usal.es (V.R.M.); idu036033@usal.es (J.N.A.); ivanmaldonadomarcos@usal.es (I.M.M.); angelavr@usal.es (Á.V.R.); alhesa@usal.es (A.H.-S.); jmhr@usal.es (J.M.H.R.); 2Servicio de Hematología, Complejo Asistencial Universitario de Salamanca, 37007 Salamanca, Spain

**Keywords:** acute lymphoblastic leukemia, next-generation sequencing (NGS), whole-genome sequencing, personalized medicine

## Abstract

Acute lymphoblastic leukemia (ALL) remains the most frequent cancer among pediatric patients and accounts for about a quarter of adult leukemias. Understanding the biological processes through which leukemogenesis originates and is maintained remains a challenge. Advances in different omics technologies have broadened our understanding and have enabled better diagnosis and management of patients by enabling the identification of biomarkers with prognostic impact, including genomic, transcriptomic, and epigenomic markers. Therefore, the aim of this review is to compare, summarize, and highlight some of the recent findings in acute lymphoblastic leukemia based on NGS applications.

## 1. Introduction

Acute lymphoblastic leukemia (ALL) is a hematological malignancy characterized by the clonal expansion of abnormal lymphoid precursors in the bone marrow, as a consequence of alterations in cell differentiation and maturation processes through the acquisition of aneuploidies, chromosomal rearrangements, and driver mutations in genes related to lymphoid cells development [1,2]. It can affect the lymph nodes, thymus, spleen, testicles, and central nervous system, and it may be derived from precursors of both T (T-ALL) or B (B-ALL) lineages [3,4]. B-ALL is the most common phenotype in both pediatric and adult patients, representing 75% and 85% of cases, respectively [5,6].

Although ALL is considered an orphan disease given that it has an age-standardized incidence rate (of 0.4 to 2.12 per 100,000 people per year) [7,8], this neoplasm still represents public health since around 6000 new diagnoses are reported each year [9]. In addition, it remains the most common childhood neoplasm, representing between 14% and 19% of all childhood cancers. It is also the second most common acute leukemia in adults [10]. Despite advances in diagnosis and the use of targeted therapies that have made it possible to achieve 5-year event-free survival rates of around 80% and 5-year survival rates approaching 90% in pediatric patients, as well as rates between 75 and 80% in adolescents and young adults (AYAs) [11], about 15% of these patients may present relapsed/refractory (R/R) disease [12,13]. Regarding adults, long-term survival decreases with age, and 5-year overall survival is estimated to be around 40% in those older than 40 years [8,14]. Therefore, a better understanding of the biological mechanisms and identifying biomarkers that can be associated with leukemogenesis remains a challenge.

ALL diagnosis requires the presence of 20% or more lymphoblasts in the bone marrow or peripheral blood, based on morphological assessment. Classification must be performed using cell morphology, immunohistochemistry, and immunophenotyping through flow cytometry. Karyotyping is still considered the “gold standard”, given that the majority of chromosomal rearrangements or aneuploidies are well characterized [6,15], may represent a strong independent prognostic factor [16], and can be used for risk assignment for treatment and therapy selection [3,17]. Nevertheless, approximately 25% of ALL cases may have a normal karyotype that could harbor high rates of submicroscopic aberrations, or cytogenetic analysis could fail [18,19,20]. In addition, around 25% to 40% of ALL patients display defined genetic abnormalities that cannot be identified by these methods [21]. In this sense, other approaches of molecular biology with higher sensitivity and specificity such as multiplex ligation-dependent probe amplification (MLPA), chromosomal microarrays (CMA), and comparative genomic hybridization (aCGH), may be also used in clinical practice [22]. Moreover, the development and implementation of next-generation sequencing (NGS) technologies for whole genome sequencing (WGS), whole exome sequencing (WES), whole transcriptome analysis (WTS), target sequencing (TGS), and epigenetic analysis have resulted in the identification of new genetic variants including point mutations, intrachromosomal amplifications, or copy number variants (CNVs) in genes such *PAX5*, *TP53*, *IKZF1*, *CDKN2A/2B,* and *RB1* in B-ALL, as well as *NOTCH1*, *FBXW7*, and W*T1* in T-ALL [23], to assess specific altered signaling pathways such as RAS or PI3K/AKT/mTOR [24], the association the presence of polymorphisms with toxicity or response to treatment in genes such as *MTHFR* (rs1801133) or *TPMT* [25,26], and also different epigenetic signatures according to phenotype and genetic subtype [27,28]. These results have allowed for a better understanding of leukemogenesis, thus providing different classifications to establish genetic subtypes associated with prognoses such as those proposed by the World Health Organization (WHO-HAEM5) or the International Consensus Classification of Myeloid Neoplasm and Acute Leukemias (ICC). In addition, the following guidelines for the management of ALL have been provided: 2024 ELN recommendations from a European expert panel for Philadelphia-negative, Philadelphia-positive, or Philadelphia-like subgroups [29] and NCCN (2024). This translates into better ALL patient management when the molecular characteristics and clinical features are integrated. In this context, this review aims to compare, synthesize, and highlight the recent insights into ALL based on NGS analysis using a narrative framework (Panel A), an opportunity for personalized medicine.
Panel A. Search strategy and selection criteriaKeywords such as “acute lymphoblastic Leukemia”, “high-throughput sequencing”, “massive sequencing”, “NGS”, “ALL risk factor”, “molecular classification”, “polymorphism”, “omics data”, and their combinations were used. Tags such as [tiab][res] or [intext: intittle] and quotation marks as search operators and (*) were used as wildcard characters or to narrow the search.The inclusion criteria were established using the following: 1. subject area and sub-areas (medicine, hematology, oncohematology, genetics), 2. type of study and publication (clinical and in vitro studies, clinical trials, observational studies, clinical practice guidelines), 3. period of analysis and language (the initial search was conducted between 2017 and 2023, and older references were also used where appropriate; only English or Spanish articles were included), and 4. availability: only open access documents were included. References include original articles, systematic reviews, and meta-analyses.The information retrieval was performed using Health in Science Descriptors (Medical Subject Headings of the National Library of Medicine of the USA-MeSH) and Hierarchical Code/thesauri, as well as databases such as Elsevier-Science Direct, Web of Science (Web of Knowledge-WoK), Scopus, and PubMed.

## 2. Next-Generation Sequencing (NGS) Applications in ALL and New Insights

In recent years, the genomic analysis of ALL in clinical practice has been performed using standard techniques that allow for the detection of the most frequent alterations defining molecular subtypes, such as ploidy alterations (hyperdiploidy or hypodiploidy), intrachromosomal amplifications, or gene fusions (*BCR*::*ABL1*, *ETV6*::*RUNX1*, *KMT2A*r) (Figure 1).

However, these methods are limited in the assessment of other cryptic rearrangements that were recently proposed for B-ALL, such as *DUX4*, *EPOR*, *ZNF384* and *MYC* rearrangements, and single-point mutations (*PAX* P80R or *IKZF1* N159Y) [30], as well as T-ALL subtypes defined by *TAL-LMO* or *LYL1* in association with gene expression signatures [31,32]. In addition, copy number variants (CNVs) and phenocopies cannot be detected with these methods. For this reason, NGS methods have been proposed to improve the characterization and detection of biomarkers that allow for better certification and management of ALL patients in the context of personalized medicine

The advances in NGS have revolutionized our understanding of leukemogenesis through the characterization of comprehensive genomic, transcriptomic, and epigenomic profiles [33,34], which provide unprecedented insight into the genetic and molecular heterogeneity of ALL. In addition, these technologies offer a powerful method for detecting a wide range of genetic variants, SNPs, insertions, deletions, and structural rearrangements and enable the identification of numerous genetic subtypes of ALL with prognostic value (Table 1). On the other hand, although the implementation of these technologies in clinical practice is promising because they allows diagnoses to be refined and could contribute to the personalization of treatment strategies in ALL patients, some limitations must be taken into consideration (Table 2).

NGS-based methods currently display variable sensitivity in detecting copy number variations (CNVs) and chromosomal rearrangements, limiting their ability to offer a fully comprehensive view of genomic alterations. As bioinformatics tools for analyzing CNV data continue to evolve, it remains essential to use NGS-based CNV analysis in combination with well-established cytogenetic methods, such as karyotyping, fluorescence in situ hybridization (FISH), SNP arrays, multiplex ligation-dependent probe amplification (MLPA), or optical genome mapping (OGM). These additional techniques provide complementary insights that address the limitations of NGS and help ensure accurate structural variant detection.

OGM is showing promise for future clinical applications, with emerging evidence supporting its utility in practice. The International Consortium for Optical Genome Mapping in Hematologic Malignancies has recently demonstrated a framework for clinical implementation, highlighting the technology’s potential to enhance diagnostic precision in hematologic malignancies [51,52]

Due to the complexity of CNV analysis, where detection sensitivity can vary based on sequencing strategies and variant sizes, NGS-based CNV detection should employ a multi-algorithmic approach. Comparing results across different bioinformatics tools, particularly when using targeted sequencing panels, can greatly enhance the accuracy of CNV identification [53,54]

## 3. New Insights into ALL Genomic Profiling and Recent Classification Based on Molecular Features

From its early stages, the classification of acute lymphoblastic leukemia (ALL) has undergone significant evolution. The first ALL classification was proposed by the French–American–British group in 1976 based on morphological and cytochemical criteria. This foundational system was later expanded by the WHO to include cytogenetic features [55]. With the advent of flow cytometry and high-throughput sequencing technologies, newer and more precise classifications have emerged. In 2022, the International Agency for Research on Cancer (IARC) released the latest version of the WHO classification of hematolymphoid tumors (WHO-HAEM5). This classification system establishes hierarchical categories that include family/class and entity/type and proposes thirteen categories for B-lymphocytic leukemia/lymphoma and two for T-lymphocytic leukemia/lymphoma [55,56]. Additionally, other groups, such as the German Multicenter Study group for Adult ALL (GMALL), the European Group for Immunological Classification of Leukemias (EGIL), the Munich Leukemia Laboratory (MLL), and the International Consensus Classification (ICC) of Myeloid Neoplasm and Acute Leukemias, have suggested incorporating additional molecular subtypes. These recommendations are based on advanced NGS applications such as WES or WTS with gene expression profile (GEP) analysis to identify phenocopies and novel fusion genes (Table 3 and Figure 2). These evolving classifications reflect the continuous advancements in genomic technologies and their profound impact on understanding and treating ALL.

## 4. Molecular Landscape of New and Emerging Genetic Subtypes of B-ALL

### 4.1. New Insights in Philadelphia ALL Patients (BCR::ABL1-Positive)

Although Philadelphia-positive ALL (Ph+) patients are typically characterized by *BCR::ABL1* fusion, the use of NGS has demonstrated that the genomic and transcriptomic profiles can be heterogeneous. Kim et al. (2020) [59] proposed two subtypes according to the *BCR::ABL1* isoform, and the cell type from which it came: the long isoform (p210) originating from a hematopoietic stem cell (HSC) that could generate a multi-lineage (multi-Ph) phenotype; and patients with short isoform (p190) and uni-lineage phenotype, from a B-cell progenitor [59]. However, these results were controversial, as recent follow-up work on residual disease shows that the p190 isoform can also be maintained in HSC with a typical B-cell phenotype [60].

Recently, Jaeseung C. Kim et al. (2023) [61] employed WES, WGS, scRNA-seq, and ATAC-seq and proposed new subtypes with prognostic value in Ph+ ALL patients. Each subtype was defined by cooperating events in genes that regulate B-cell development and not by the cell of origin where the event occurs. C1 (from an early B-cell progenitor, “Early-Pro”), C2 (from an intermediate B-cell progenitor, “Inter-Pro”), C3 (late B-cell progenitor, “Late-Pro”). C1 presented higher leukocyte counts at diagnosis and was characterized by aberrant expression of stem cell and myeloid genes, such as *KIT*, *CSF3R*, and *MECOM*, as well as an increased expression of genes related to innate immunity, such as *TNFα.* Group C2 showed differential expression of both myeloid (*CSF2RA* and *CSF1R*) and lymphoid (*MS4A1/CD20* and *IL7R*) genes, as well as genes related to the ERK signaling and unfolded protein response (UPR) pathways. Group C3 showed the highest expression of genes associated with B-cell differentiation, such as *IL7R*, *MS4A1*, *BACH2,* and *TCL1A*, and cell cycle regulatory genes. Analysis of IGH heavy-chain expression was similar in all groups, while the expression of rearranged light chains (IGK or IGL) was more frequent in the C3 group, which also showed higher expression of B-cell factors such as *EBF1* and *PAX5*, suggesting a later differentiation block compared with the other groups [61].

Another relevant fact among Ph+ leukemias is the *IKZF1* deletions and other cooperative alterations in genes related to B-cell differentiation such as *PAX5*, *CDKN2A/B* and *EBF1*, in addition to other genes such as B*TG1*, *RB1*, *EBF1*, *MEF2C*, *RUNX1*, *SLX4IP*, *HBS1L*, *TSC22D1*, *FOXP1*, *LEF1*, *ETV6*, *TCF3* and *TCF4* [62]. Jaeseung C. Kim et al. (2023) conducted a WGS analysis and described new recurrent alterations in *CBWD2*, *GPN3/FAM216A*, *PRKAR2B* (without monosomy 7), *MAP3K2*, *MIR181A1HG/MIR181B1,* and *FOXP1*. C1 leukemias present a higher proportion of alterations, such as monosomy 7 or 1q gain, and *HBS1L* deletions; *EBF*1 and *RUNX1* variants are restricted to this group. However, C2 and C3 present a higher number of alterations in genes such as *PAX5*, *CDKN2A*, and *SLX4IP*, as well as hyperdiploidy. In addition, although *IKZF1* alterations were present in all subtypes, biallelic losses and the dominant negative Ik6 deletion of *IKZF1* were more frequent in C2, suggesting that complete inactivation *IKZF1* drives the Inter-Pro subtype. The C3 group had higher frequencies of *PAX5* deletions and homozygous deletions of *CDKN2A/B* and *RB1*, which was consistent with a posterior block in B-cell differentiation.

Regarding the immunophenotype and molecular response, differences were also found between the groups. C1 presented myeloid antigens, CD13, CD33, and cytoplasmic myeloperoxidase (cyMPO), CD7, and the highest expression of CD34, while the C3 group displayed the highest levels of CD10, CD19, and CD20. Fewer C1 and C2 patients achieved a major molecular response (MMR, log reduction ≥3, transcript ≤0.1%) compared with C3 patients, who had a deep molecular response (log reduction ≥4); consequently, they had higher overall survival (OS) and event-free survival (EFS) rates. On the other hand, the authors also propose that prognostic staging should take the occurrence of mutations inside or outside the BCR::ABL kinase domain into account. C1 patients exhibit an enrichment of BCR-ABL1-signaling pathways and its target STAT5, as well as increased phosphorylation of STAT5. This has been described as a mechanism of resistance to TKI in the absence of kinase domain mutations. In contrast, Lorenz Bastian et al. (2024) [63] proposed two main clusters within ALL Ph+ based on transcriptomic and genomic profiling, as well as ATAC-Seq analysis. The authors found 331 differentially expressed genes that allowed for the establishment of complex transcriptomic signatures for the two main and four subclusters: multilineage (del*HBS1L* or del7) and lymphoid (homozygous *IKZF1* deletion, *CDKN2A/PAX5,* and/or hyperdiploidy with strong enrichment for the Hippo and Hedgehog pathways, respectively). HBS1L deletion harbored the same breakpoints in all samples (chr6:135,044,863-135,116,862; GRCh38hg38), including the promoter and exons 1 to 2. Patients showed an increased *HBS1L* gene expression but reduced usage of exon 1 to 3. Long-read RNA-Seq confirmed a previously undescribed *HBS1L* transcript (HBS1Lalt) in delHBS1L and del7 cases initiated from a putative transcription start side in HBS1L intron 3 [63].

Taken together, the results suggest DNA- or RNA-based risk stratification may have clinical utility. In addition, studies should be expanded to conclude on the association of the phenotype with other clinical variables. Jaeseung C. Kim et al. (2023) indicate that subtypes are not associated with age, sex, or the BCR-ABL1 isoform (p190 or p210). In comparison, Lorenz Bastian et al. (2024) found a predominance of the *CDKN2A/PAX5* subgroup in pediatric patients despite the equal distribution of multilineage and lymphoid cases in children and AYAs. Adult patients were more frequently classified as multilineage with an increase in del7 cases. In addition, patients in the multilineage group more frequently presented major breaks in *BCR::ABL1,* with a significant increase in BCR exon14 involvement in del*HBS1L*. Moreover, white blood cell counts at diagnosis also differed between the new *BCR::ABL1* groups (highest in del*HBS1L* and *CDKN2A/PAX5*), indicating that the new group definitions also represent distinct clinical profiles.

### 4.2. Phenocopies in B-ALL

#### 4.2.1. BCR::ABL1-like (Ph-like)

This subgroup presents a gene expression profile similar to that of Philadelphia-positive patients; it occurs at all ages, although it is less common in children (<10%), and its prevalence increases to 30% in adolescents. Comprehensive NGS (WTS and WGS simultaneously) is the best approach to identifying this subtype. It is commonly characterized by *CRLF2* rearrangements (around 50% of cases) and *IKZF1* deletions. It is associated with adverse clinical features, such as a high white blood cell count, a high minimal residual disease level after induction therapy, and a high relapse rate [64]. Thus, Arber et al. (2022) highlighted the importance of distinguishing molecular subtypes among *BCR::ABL1-like* (Philadelphia-like, Ph-like) patients, considering that some patients could benefit from specific therapies [65]. These patients can be grouped as follows: (i) The ABL class, which includes 5′ *ABL1* fusions (with *CENPC*, *ETV6*, *FOXP1*, *LSM14A*, *NUP153*, *NUP214*, *RANBP2*, *RCSD1*, *SFPQ*, *SNX1*, *SNX2*, *SPTNA1*, or *ZMIZ1)*, *ABL2* fusion (with *PAG1*, *RCSD1*, or *ZC3HAV1)*; *CSF1R* fusions (with *MEF2D*, *SSBP2* or *TBL1XR1)*, *PDGFRB f*usions (with *ATF7IP*, *EBF1*, *ETV6*, *SNX29*, *SSBP2*, *TNIP1*, *ZEB2*, or *ZMYND8)*; and *PDGFRA::IP1L1* [66,67]. (ii) Altered JAK-STAT signaling, with *JAK1*, *JAK2*, *IL7R*, and *SH2B3* mutations or *EPOR* rearrangements [68,69]. J*AK2* gain-of-function mutations may occur in association with *CRLF2* overexpression in around 50% of cases. The latter occurs as a consequence of translocation, leading to *IGH-CRLF2* juxtaposition or a *PAR* deletion (PAR1, Xp22.33/Yp11.2) and generating *P2RY8-CRLF2* fusion [70,71]. The third grouping is (iii) *BCR*::*ABL1*-like not otherwise specified (NOS), with genetic fusions or mutations involving genes such as *ETV6*, *BLNK*, *DGKH*, *PTK2B*, *FLT3*, *FGFR1*, and *TYK2*. In total, 1% of Ph-like cases harbor an *ETV6::NTRK3* fusion that is associated with aggressive ALL [72,73]. On the other hand, CNV analysis of *IKZF*, *CDKN2A*/*B*, *PAX5*, *PAR1*, and *ERG* in *BCR*::*ABL1*-like patients has been proposed as it could allow for the identification of IKZF1^plus^ or IKZF1-deleted profiles that may have prognostic significance, and this could guide more personalized therapeutic strategies to improve outcomes in this group [54].

#### 4.2.2. ETV6::RUNX1-like

This genotype occurs in approximately 2–3% of cases in children and in less than 1% of cases in AYAs and adults [74,75]. This subgroup harbors *ETV6* aberrations and is characterized by a CD27-positive/CD44-low to -negative immunophenotype and a similar gene expression profile to that of the *ETV6::RUNX1* subtype. This genotype may harbor *IKZF1*, *TCF3*, *ERG*, or *ARPP21* CNVs [76]; gene fusions such as *IKZF1::ETV6*, *ETV6::ELMO1*, *TCF3::FLI1,* or *FUS::ERG* [77,78]; and biallelic inactivation of *ETV6* by de novo deletion or in pediatric patients that harbor a germline *ETV6* mutations with subsequent somatic alterations of the second *ETV6* allele. This biallelic ETV6 alteration is central in leukemogenesis [76,79]. On the other hand, Dongfeng Chen and colleagues (2020) proposed a RAG1 signature belonging to the *ETV6::RUNX1*-like subtype, which shows high RAG1 mRNA levels and over-expression of *ETV6* target genes as well as *BIRC7*, *WBP1L*, *CLIC5*, and *ANGPTL2* [75]. The prognostic value is unclear. Some studies have shown an average 5-year EFS of 66.7%, 5-year OS > 87%, and an MRD negative at the end of consolidation (EOC) of >86%, suggesting that this subtype may benefit from higher intensity therapy [2,80]. However, Jeha S et al. (2021) suggest that this genotype confers an unfavorable prognosis in children due to the high levels of MRD and worse event-free survival rates [81].

#### 4.2.3. ZNF384r-like (Provisional Entity)

This subtype was characterized using RNA-seq, gene expression profiles (GEPs), and a machine learning classifier (ALLSorts) [82]. This cluster presents a similar ZNF384r expression profile without 5′ canonical changes in *ZNF384*, including indels such as p.393_397KHNPP>RAG, p.483_484CS>YFEWDPND, and p.481_516>SFQDYCACQ that lead to a disruption in the C-terminal [83]. Likewise, a *ZNF362(ZNF384* paralog) rearrangement has been identified in this subgroup. Similarly to ZNF384r leukemias, this entity has a predominant BCP immunophenotype. Myeloid markers (CD13/CD33) are often expressed, CD10 is weakly expressed, and some patients may also have aberrant expression of CD66c [84,85]. The prognostic value is yet unknown.

#### 4.2.4. KMT2Ar-like (Provisional Entity)

This subgroup was proposed by Amy S. Duffield et al. (2023) based on WTS and gene-expression-profiling data from 2041 leukemia samples collected at diagnosis. They found some cases (<1%) with a differential profile between KMT2Ar cases. They proposed that these patients may harbor HOXA fusion. As with ZNF384r-like, the prognostic value is currently unknown [58].

### 4.3. Novel Gene Fusions in B-ALL

#### 4.3.1. MYC-Rearranged

This rearrangement may occur around 2% to 5% cases B-ALL, and it is rare in T-ALL (1% in children, and it is under-recognized in adults) [86,87]. It is associated with an extremely aggressive syndrome characterized by hyperleukocytosis, rapid neurological progression, and poor response to chemotherapy [88,89]. However, recent studies show that the combination of intensive chemotherapy and rituximab can improve the prognosis, allowing for the cure of about 50% of this subtype [90,91]. Usually, these patients present MYC overexpression due to t(8;14)(q24;q11), which places the *MYC* gene under the control of strong T-cell-responsive enhancer elements (*TCRA* or *TCRD*) [92] or other partner genes, including *IGL*, *IGK*, *BCL2* (<3%) [16,93,94], and *BCL6* (<1%) [56,58]. The prognostic significance varies according to the partner gene. Recent studies suggest that MYC::IGH fusion could confer good prognosis with high-dose methotrexate and cyclophosphamide [16]. In contrast, *MYC::BCL2* can be associated with a worse outcome [16].

#### 4.3.2. IGH Fusions, t(v;14q32)

These fusions represent around 5% of all B-ALL patients, and the rate is higher in adults. *IGH* fusions are usually associated with poor prognosis independent of the partners genes, including *BCL2*, *BCL6*, *ID4*, *MYC*, *CEBPA*, *CEBPB*, *IGF2BP1*, *IGK*, *EPOR*, *LHX4*, *IL-3,* and *CRLF2* [16,86,95]. Nevertheless, specific cases, such as IGH::BCL2, should be rigorously analyzed since low levels of circulating t(14;18)-positive cells have been found in approximately 50–70% of healthy individuals who never develop ALL [80,81]. *IGH::CRLF2* fusion is found almost exclusively in the subset of Ph-like ALL (~50%) [94,96].

#### 4.3.3. TCF3-Rearranged (19p13.3)

This accounts for about 4–6% of all pediatric B-ALL cases and is less frequent in adults (<3%) [2,16,97]. *TCF3*r is overrepresented in hyperdiploidy and *ETV6::RUNX1* ALL cases (~8%) [94], and it may be accompanied by *IKZF1* deletions [2]. The phenotype is characterized by moderate CD10 and CD19, strong expression of CD9, low to negative CD34, and at least a partial absence of CD20. This may be associated with *PAX5* or *VPREB1* deletions or mutations in Ras pathway genes (*NRAS*, *KRAS or PTPN11*) [98,99,100]. Although der(19)t(1;19) chromosomes can be detected using conventional cytogenetics or FISH, using other NGS strategies has allowed for the characterization of additional TCF3 partner genes. The most common rearrangement is *TCF3::PBX1* fusion involving TCF3 exons 1–16 and exons 3–9 of PBX1. However, in up to 10% of cases, there are alternative break-points [101]. Other partners genes include *ZNF384* (12p13) [102], *FLI1* (11q24.3), *TEF* (22q13.2), and HLF (17q22). HLF is not normally transcribed in lymphoid cells, but fusion leads to the leukemic cells becoming similar to stem cells [100,101,103]. In addition, Fischer et al. found an association of TCF3/HLF with *PAX5* haploinsufficiency and enrichment in stem cell and myeloid expression signatures [104,105]. Even though it is present in <1% of ALL cases, this genotype was recently included in WHO-HAEM5. In addition, it is associated with poor prognosis, early relapse, and death within two years of diagnosis [106,107]. Another relevant aspect is the consideration that these patients exhibit a high CD19 expression, and immunotherapy, stem cell transplantation, and CAR T-cell therapy may hold promise [108].

#### 4.3.4. DUX4-Rearranged

DUX4r may be present in 3% to 10% of the cases of B-ALL, with a higher proportion of cases in AYAs; it is associated with a favorable prognosis, despite the presence of high MRD levels [109,110]. These cases have a distinct CD2-positive and CD13-, CD34-, and CD38-overexpression immunophenotype [111], as well as a GEP characterized by *DUX4* and *ERG* deregulation. The main rearrangement is a linkage between *IGH* and *DUX4* (4q and 10q D4Z4 repeat domains), but t(4;21)(q35;q22) DUX4/ERG also may occur [111,112]. *DUX4* over-expression can be detected using IHC, but according to the diagnosis criteria in WHO-HAEM5, RNA or DNA sequencing must be used to identify other cryptic rearrangements and simultaneous co-lesions, such as *NRAS-* or *KRAS*-activating mutations, *IKZF1*, *CDKN2A/B*, or *ERG* deletions or *ERG* loss of function (ERGalt). ERGalt is generated by aberrant DUX4 binding to intragenic region *ERG*, which leads to an *ERG* isoform that retains the DNA-binding and transactivation domains inhibiting the *ERG* wild-type allele with a dominant negative effect [76,113]. Moreover, genomic studies have revealed that around 50% to 70% of cases display intragenic ERG deletions, a genomic aberration that is practically nonexistent in other B-ALL cases [114].

#### 4.3.5. ZNF384-Rearranged (12p13.31)

This represents 3% of all pediatric B-ALL cases and around 7–10% of cases in AYAs. It has been proposed that its prognostic value is controversial [115]. Ya-Zhen Qin et al. (2021) found significantly higher 3-year relapse-free survival (RFS) and less overall survival (OS) compared with those without [116]. In contrast, Zhu et al. (2023) proposed that patients commonly have a good prognosis, but this might depend on the fusion partner gene [117]. This subtype exhibits abnormal expression myeloid markers CD13 and/or CD33. However, the typical marker of an immature B lineage, CD10, is only weakly expressed in some of them. The most common fusion partners are *EP300/ZNF384*, *CREBBP/ZNF384*, and *TCF3/ZNF384,* but *TAF15*, *EWSR1*, *ARIDIB*, *BMP2K*, *CLTC*, *NIPBL*, *SMARCA2*, and *SYNRG* have also been reported [85] [115,117,118]. ZNF384r carries other co-lesions, such as Ras pathway gene mutations; *FLT3* overexpression; *CLCF1* and *BTLA* upregulation; JAK-STAT pathway activation due to *GATA3*, *CEBPA,* and *CEBPB* over-expression [119]; and *LEF1*, *EBF1*, *CDKN2A*, *FBXW7*, *CDKN2A/2B*, *IKZF1*, *PAX5*, *RUNX1*, and *ETV6* deletions [2]. *ZNF384*r should be detected using WTS or DNA-seq while taking into account that specific genomic profiles could benefit from targeted therapies such as histone deacetylase or JAK-STAT pathway inhibitors [120,121].

#### 4.3.6. MEF2D-Rearranged (1q22)

The incidence ranges from 2% to 5% of all B-ALL cases, and it is more frequent in children and AYAs. The immunophenotype is characterized by CD10-negative status, CD38-positive status, and, often, aberrant CD5 expression [122,123]. *MEF2D* rearrangements preserve the MEF2D DNA-binding domain, resulting in enhanced transcriptional activity, including that of 3′ partner genes such as *BCL9*, *HNRNPUL1*, *HNRNPM*, *DAZAP1*, *FOXJ2*, *HNRNPH1*, *STAT6*, *SF1R*, *SS18,* and *CSF1R* [124,125]. Those fusions are usually mutually exclusive with the known risk stratification of chromosomal translocations (at least at the cytogenetic level), except for *MEF2D::CSF1R,* which display a Ph-like profile [78]. Additional genetic alterations include copy number oscillations involving multiple chromosomes, suggesting chromothripsis, *IKZF1* and *CDKN2A/B* CNVs, dysregulation of RNA stability, *NRAS* mutations [123,124,126], or *GATA3* and *HDAC9* overexpression. This suggests that these patients could benefit from histone deacetylase inhibitors such as Panobinostat [2,16,123]. The clinical features are still not well defined. Nevertheless, it has been considered as a high-risk molecular subtype, given that patients may have elevated white blood cells, ineffective stem cell transplantation, and relapse rates around 53.3% [125,127,128]. Although FISH probes exist for the detection of fusions such as *MEF2D::BCL9*, other techniques, such as WGS, WTS, or OGM, could be used to identify other less frequent or novel rearrangements.

#### 4.3.7. NUTM1-Rearranged (15q14)

This has been reported in about 3% of B-ALL cases, and it is more common in infants (<1 year) than in children (0.4–0.9%), while it is rare in adults [92,118], and it is associated with a good prognosis. According to Judith M. Boer et al. (2021), the 4-year event-free and overall survival rates were 92.1% and 100%, respectively, and 91.1% of patients had an MR of <0.05% [129]. This rearrangement might involve many 5′ partners, including *BRD9*, *BRD4*, *IKZF1*, *CUX1*, *AFF1*, *ZNF618*, *ACIN1*, and *SLC12A6*, increased aberrant *NUTM1* expression, and altered global chromatin acetylation. This fact might confer histone deacetylase inhibitors with sensitivity [129,130,131]. The downstream effects of *NUTM1*r include the down-regulation of the NOTCH and Hedgehog signaling pathways, proto-oncogene *BMI1* up-regulation, and high expression of clustered genes on chromosome band 10p12.31-12.2, including *HOXA* gene cluster. In fact, in 2021, two biological subgroups within the NUTM1r subtype were suggested: (1) HOXA9-positive, with a limited number of partners (*ACIN1*, *CUX1*, *BRD9*, and *AFF1*), which is more prevalent in infants of <9 months of age and was reported in association with *CRLF2*r; (2) HOXA9-negative, which occurs in infants approaching 1 year of age. Almost half of all NUTM1r pediatric cases occur in this subgroup. Until now, no other somatic sentinel alterations have been reported, suggesting that NUTM1r could be a possible oncogenic driver event [131,132,133].

#### 4.3.8. UBTF::ATXN7L3/PAN3, CDX2 (Also Known as “CDX2/UBTF”)

This entity has been reported in approximately 1% of ALL cases and is more common in AYAs and females. It is characterized by a CD10-negative and IgM-positive immunophenotype. This genotype is featured by two co-occurring genetic lesions: a cryptic deletion at the 17q21.31 locus involving loss of the last four UBTF exons (exon 18 to exon 21) and an intergenic sequence, generating a chimeric *UBTF::ATXN7L3* protein [134]; a deletion of approximately 41Kb (common and minimal-size deletion) encompassing the intergenic region between *FLT3* and *PAN3*, resulting in monoallelic ectopic expression of *CDX2* [135]. Nevertheless, it must be analyzed rigorously that *CDX2* overexpression and other 13q12.2 microdeletions that deregulate *FLT3* have been reported in B-ALL [136]. Moreover, individual analysis of *CDX2* has limited applicability for the identification of this molecular subtype, but *CDX2* overexpression has been reported in other leukemias associated with aberrant *HOX* gene expression [137,138,139]. CDX2/UBTF cases may harbor additional genetic alterations, including 1q gain and *CXCR4-*activating mutations; *PAX5*, *VPREB1*, and/or *IKZF1* deletions; and *PRY2* and *CD9* overexpression. Furthermore, a ChiP-seq analysis showed that 13q12 region deletion alters DNA topology, and that the epigenetic pattern can be altered, as H3K4me3 and H3K27ac spikes have been observed in the *CDX2*, *FLT3,* and *PAN3* promoters [135]. Regarding risk, this subtype is classified as having poor prognosis. These patients have a higher risk of failing the first induction course and postinduction MRD, resulting in a significantly higher cumulative relapse incidence [134].

### 4.4. Point Mutations

#### 4.4.1. PAX5^P80R^

*PAX5* mutations used to be considered as secondary events in the leukemogenesis process. However, the identification of a group with *PAX5* mutation (P80R) without associations with any other typical features, and a different expression profile according to WES, allowed for the establishment of this new ALL subtype [37]. P80R is the most common *PAX5* mutation; it is observed in around 2–5% of B-ALL cases and is more frequent in adults [140,141]. Some cases may present biallelic inactivation of *PAX5* due to wild-type allele deletion, homozygous *CDKN2A/B* or *IKZF1* deletion, and hotspot-activating mutations in RAS-signaling genes. GEP studies also reveal a JAK-STAT-signaling alteration [37,142,143]. As with patients with the *DUX4*r and *ZNF384*r subtypes, these patients frequently undergo a monocytic switch with a CD2-positive, CD20-low, CD45-high, CD13-negative, CD33-positive, and CD10-high profile [36,83,141]. The prognostic value is variable according to age. Some authors consider it to be an intermediate risk in children and to have a good prognosis for adults [141,142].

#### 4.4.2. IKZF1^N159Y^

This represents approximately 1% of ALL cases and is characterized by a missense mutation that affects the DNA-binding domain [36,118]. In general, somatic *IKZF1* alterations are associated with high-risk B-ALL with poor response to therapy, but the prognostic impact is still controversial [144,145]. *IKZF1*^N159Y^ is associated with gain of chromosome 21, *KRAS* mutations, B-cell receptor downregulation, JAK-STAT-signaling alteration (*FLT3*, *FLT4*, and *STAT5A*) [118], up-regulation of the *IKZF1*-interacting gene *YAP1* (a transcriptional co-activator that can drive KRAS-induced transformation by rescuing cell viability in KRAS-dependent cells), *ARHGEF28*, and overexpression of the chromatin remodeler *SALL1* over-expression [36,84]. Previous studies have reported that *SALL1* can recruit histone deacetylase (HDAC) to mediate transcriptional repression. Its promoter is frequently methylated in BCP ALL [35]. On the other hand, partial *IKZF1* partial tandem duplications (IKZF1-PTD) ranging in size from 344 bp to 396 bp and span the N159Y variant have been observed in exon 5 [146].

### 4.5. Other Emerging B-ALL Subtypes

#### 4.5.1. PAX5-Altered

*PAX5* is considered a master regulator of B cell development and maintenance and is the most commonly altered gene in ALL, as it is detected in approximately one-third of cases [147]. It is associated with poor prognosis, but other clinical variables must be taken into consideration [142]. This molecular subtype includes CNVs (whole-gene, partial, or intragenic regions), translocations, different point mutations to P80R that generate haploinsufficiency, and combinations thereof that may result in biallelic inactivation [143].

Monoallelic deletions of *PAX5* (focal deletion, loss of gene regions of 9p, or chr9 monosomy) are the most common PAX5 variants in B-ALL and are considered a secondary or late event. This has been reported in more than 50% of *BCR::ABL1* cases, about 20% of *TCF3::PBX1* patients, and as a co-lesion in Ph-like [141]. It is often associated with complete *CDKN2A/B* loss; *STAT5* activation; IL-7 pathway hyperactivation; *RB1*, *BTG1*, *IKZF1*, *VPREB1,* and *BTLA* deletions; and/or epigenetic alterations (e.g., *KDM6A*, *KMT2A*) [142]. *PAX5 fusions* are less common (2%-3% of B-ALL cases) but usually appear in cases with normal karyotypes; therefore, they are considered a founding effect in the process of leukemogenesis. The majority of *PAX5* rearrangements result in chimeric genes that retain the DNA-binding paired box domain and nuclear localization domain. PAX5 3′ partner genes include *ETV6*, *JAK2*, *ZCCHC7*, *NOL4L*, and *AUTS2*, in addition to, less commonly, *ZNF521*, *FOXP1*, *DACH1*, *NCOA5*, *ELN*, *POM121*, *PML*, *MLLT3*, *CHFR*, *SOX5*, and *POM121C.* As a result, these fusion proteins may act as dominant negative inhibitor [148]. *PAX5* alterations also include *intragenic amplifications* (PAX5^AMP^) and *germline mutations.* The incidence PAX5-ITD is around 0.5–1.4% in B-ALL and consist in an approximately 4–5 copies in *PAX5* exons 2 to 5. The functional role of PAX5-iAmp should be further investigated [149,150]. The somatic variants include G183S, R38H, and R140L, as they have been shown to predispose one to familial ALL-B and to cause, at the functional level, loss of B-cell subtypes and immune deficiency. In fact, the ICC considers that germline alterations in PAX5 constitute a distinct entity [140].

#### 4.5.2. Non-Ph-like CRLF2r and “Double-Hit” BCL2/MYC (IGH/BCL2 and 8q24/MYC Rearrangement)

This genotype occurs in approximately 5% of B-ALL cases and is associated with alterations in the JAK/STAT or PI3K/mTOR pathways, high relapse frequency, and, therefore, poorer prognosis [64]. *BCL2/MYC* is a rare event, but the characterization of *MYC*r is relevant, given that the fusion partner gene could define the role of MYC and its level of expression during leukemogenesis, increasing early-relapse risk and causing a poor prognosis [89,151].

## 5. New Insights into Genomic Profiling of T-ALL

Establishing molecular subtypes in T-lineage acute leukemia is more complex. Based on immunophenotypic criteria, a “gray zone” is found, given that some cases do not have clear lineage-specific antigens [152]. Initially, four main T-ALL subtypes were proposed: mature (sCD3 positive), cortical (CD1a+), pre-T (cCD3+ plus CD2+, CD5+, and/or CD8+), and pro-T ALL (cCD3+ and CD7+); however, with the integration with RNA-microarrays, the T-ALL subtypes began to be better characterized [153,154,155].

The most common alterations in T-ALL includes the TAL gene family (*TAL1*, *TAL2*, and *LYL1* proto-oncogenes) and LMO family rearrangements [156] (Figure 3). In children, these alterations account for approximately 30%, and in adults, the incidence is lower (~12%). These rearrangements may be accompanied by *CASP8AP2* (20%) or *PHF6* mutations and *PTEN* deletions. Other common alterations include HOXA (*HOXA10*, *HOXA9*) rearrangements or overexpression (20–30% of cases) [157,158], accompanied by *KMT2A*, *MYB,* and *SP1* rearrangements; *PICALM::MLLT10* or *SET::NUMP214* fusion; or *ETV6*, *CNOT3*, *EZH2*, *JAK3*, *and STAT5B* mutations [159,160]. The mutational T-ALL landscape also includes *CDKN2AB*, *TLX1*, *NKX1*, and *KMT2A* deletions; *ABL1* rearrangements; and *IL7R*, *JAK3*, *JAK1*, *WT1*, *FBXW7* [99], PTEN, RPL10 [155], or *NOTCH* (9q34.3) mutations [161,162]. This variant is associated with *CDKN2A* and *FBXW7* deletions and occurs in approximately 50% to 60% of pediatric T-ALL cases. These deletions are associated with poorer prognosis [163]. *TLX3* rearrangements are also common in T-ALL. However, more studies are needed to compare the clinical and prognostic implications, as the results are inconsistent [164].

According to WHO-HAEM5, there are only two T-ALL groups that can be clearly differentiated. T-ALL NOS, and T-cell precursor early acute lymphoblastic leukemia (ETP-ALL) are characterized by an immunophenotype with negative or absent CD1a and CD8 (<5% positive cells), negative or low CD5 (<75% blast cells), and expression of one or more myeloid (CD11b, CD13, CD33, CD117) or stem cell markers (CD34, HLA-DR) [155,165,166]. The genomic and transcriptomic profiles are characterized by activating mutations in genes regulating cytokine receptor signaling (*RAS BRAF*, *NRAS*, *KRAS*, *FLT3*, *JAK1*, *JAK3*, and *SH2B3*) (~67% of cases), inactivating lesions that affect hematopoietic development (*GATA3*, *ETV6*, *RUNX1*, *IKZF1*, and *EP300*), or histone-modifying genes (*EZH2*, *EED*, *SUZ12*, *SETD2* and *EP300*), as well as *MLH3*, *SUZ12*, *PHF6*, *WT1*, *DNM2*, *ECT2L*, or *REL* mutations [159,160,166]. Other relevant mutations in genes such as *RELN*, *MIR1297*, and *IL7R* were proposed by St Jude Children’s Research Hospital in the Cancer Genome Project by performing WGS for matched leukemic and normal DNA from pediatric ETP-ALL patients. Two hotspots of missense and in-frame insertion–deletion mutations (IL241-242 and VA253-254) of *IL7R* were reported [167].

## 6. Conclusions

Recent advances in NGS and their translation to clinical practice have improved the characterization of the ALL patient mutational landscape, risk stratification, and management. This review highlights new and emerging subtypes of ALL included in recent system classifications, such as WHO-HAEM5 (2022) and ICC (2022), and their prognostic value based on their molecular features, including somatic mutations, gene fusion, epigenetic markers, transcriptional profiles, and alterations in molecular pathways that allow for defining phenocopies.

NGS technologies, such as whole-genome sequencing, targeted sequencing, transcriptome analysis, chromatin immunoprecipitation followed by sequencing (ChIP-seq), ATAC-seq, among others, have proven to be powerful tools for advancing the knowledge of ALL. In the research field, these technologies have allowed for genomic characterization even at the single-cells level to identify driver mutations related to disease development in high-risk subtypes such as *KMT2A*r [168,169], or genetic alterations relevant to disease progression and treatment response in genes such as *TTN*, *FLT3*, *TP53*, *MUC16*, and *EPPK1* in B-ALL, and *NOTCH1*, *FBXW7*, *TTN*, *MUC16*, and *PHF6* in T-ALL [170]. In this sense, the prognostic value of NGS in ALL also lies in its potential to improve patient characterization at diagnosis and provide insight into the genetic heterogeneity of the disease, including the identification of alterations in genes such as *IKZF1*, *TP53*, *CRLF2*, and *JAK2*, which are associated with poor prognosis in B-ALL, to detect differential gene expression profiles or epigenetic patterns that may modulate leukemogenesis and treatment response, such as *NOTCH1* in T-ALL, to establish clonal heterogeneity and trace evolutionary trajectories that are useful for identifying alterations associated with relapse risk [171,172], as well as to characterize of differential transcriptomic profiles to identify novel risk subgroups.

Due to their impact, some NGS applications have been transferred to the clinical setting to evaluate the minimal residual disease and identify high-risk variants or subtypes with greater precision, and they have been incorporated into guidelines or recommendations, such as those of the European Leukemia Network-ELN (2022), and clinical trials, such as the PETHEMA protocol (ALL-2019), SEHOP/PETHEMA, and other protocols evaluating responses to immunotherapy. Clinical services in referral hospitals have adopted the use of technologies due to their ability to improve diagnostic accuracy, risk stratification, and minimal residual disease (MRD) as a critical factor in disease monitoring due to their higher sensitivity (factor 1 × 10^−6^) compared to polymerase chain reaction (PCR) or flow cytometry assays. In clinical practice, the use of the NGS-based ClonoSEQ assay has improved the MDR detection, even using peripheral blood samples. This assay was the first NGS-based test approved by the FDA for ALL, allowing for deep response measurement, early relapse detection, and broader genetic analysis to identify emerging mutations that may contribute to treatment resistance [173]. In addition, it is used for pre- and post-transplant monitoring in high-risk patients. This application has been included in some recommendations, such as NCCN (2024) and European LeukemiaNet recommendations-ELN (2022) [22,29]. However, other guidelines, such as the ESMO Clinical Practice Guideline (2024), highlight some considerations on the MDR detection by NGS among high-risk genetic subtypes such as Ph+, suggesting that real-time quantitative PCR for BCR::ABL and PCR for Ig/TR rearrangements are better, and NGS is useful for the delineation of CD19 clones prior to CAR-T therapy and response assessment to these therapies [174]. In addition, the prognostic value of MDR detection by NGS is also being evaluated in clinical trials, with different approaches, including evaluating the efficacy and safety of treatment schemes such as combination chemotherapy with CAR-T cells in B-ALL patients (NCT06481241), testing and monitoring B-Cell recovery to guide management after Chimeric Antigen Receptor T (CART) cell administration, and monitoring B-ALL relapse in children and young adults with B-ALL (NCT05621291).

NGS techniques are also being used in clinical trials for targeted and personalized treatments. The identification of specific mutations by NGS allows the use of targeted therapies in patients with mutations in *ABL1* or *JAK2*, improving response rates and reducing side effects compared to conventional chemotherapy. In this sense, NGS is useful for treatment decision making and contributes to prognosis. Other applications include the identification of phenocopies as Ph-like in B-ALL and their prognostic assessment (NCT02580981), the assessment of chimerism and relapse post Bone Marrow/Hematopoietic cell transplant (HCT) (NCT04635384), which analyzes SNP loci to quantify donor and recipient cells by measuring genomic DNA prior to transplantation, and the characterization of the mutational landscape or transcriptional profiles in B-ALL, such as the genetic study of familial Acute Lymphoblastic Leukemia (NCT03067584) to identify gene variants that are inherited and influence the risk of developing acute lymphoblastic leukemia, including novel SNPs as a risk. In T-ALL, clinical trials are also underway to evaluate the safety and efficacy of multiple base-edited allogeneic Anti-CD7 CAR-T cells in relapsed/refractory patients (NCT05885464), or to assess the effect of treatment as a prognostic HDACi Chidamide targeted therapy for adult T-LBL/ALL (NCT03564704).

Another concrete application in clinical practice is multicenter studies aimed at standardizing NGS protocols for the management of ALL patients at diagnosis and treatment. The inclusion of targeted panel sequencing in protocols, such as those of the PETHEMA group, and, in particular, the assessment of *TP53* status in B-ALL, and *NOTCH1*, *KRAS*, *NRAS*, and *FBXW7* in T-ALL (NCT04179929), has made it possible to evaluate mutational profiles and make decisions for the benefit of patients. Hospitals and research institutions have developed risk classifiers based on NGS data that allow clinicians to tailor treatments based on each patient’s molecular profile, an increasingly relevant approach as precision medicine advances in hematologic cancers. Advances in bioinformatics have also enabled the development of algorithms such as ALLsort based on RNAseq data that allow better classification of ALL patients [82]. In addition, the technical and clinical validation of TGS to improve the molecular characterization of ALL patients based on mutation profiles and CNVs. Others panel such as Archer™ FUSIONPlex™ Leukemia panel also allows for the identification of fusions and gene expression profiling, as well as splicing, exon-skipping, SNVs, indels, and others.

Currently, NGS methods are playing an increasingly important role in ALL both in genotyping and prognostic evaluation. While NGS effectively detects a broad range of genetic alterations, it remains complementary to traditional cytogenetic methods due to limitations in identifying certain structural variants and copy number variations (CNVs). The use of NGS technologies with gold-standard techniques is the best alternative for the management of ALL patients in the context of personalized medicine. A combined approach in which NGS and OGM are used alongside established cytogenetic techniques, such as karyotyping, fluorescence in situ hybridization (FISH), and SNP arrays, to enhance diagnostic precision and prognostic assessment. This combination approach is considered the current gold standard in ALL genotyping, providing a comprehensive understanding of the genomic landscape essential for accurate risk stratification and personalized treatment planning.

Finally, all the advances and applications mentioned in this review highlight the important prognostic and practical impact of NGS in the management of ALL patients, making it a useful cornerstone for treatment personalization and suggesting that its incorporation into routine clinical practice can significantly optimize outcomes in patients with ALL in the context of personalized medicine.

## Figures and Tables

**Figure 1 cancers-16-03965-f001:**
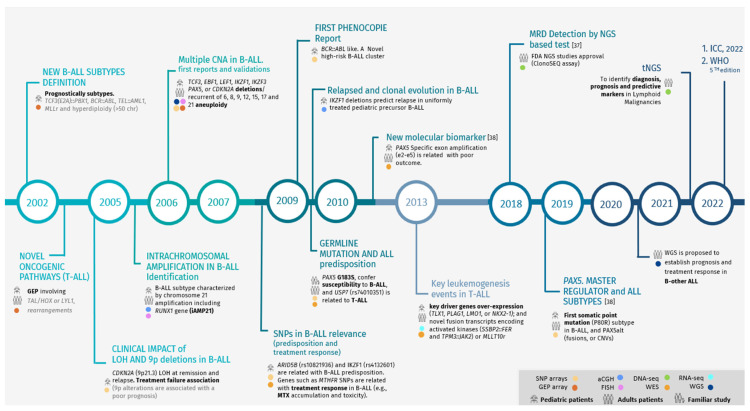
Milestones in ALL based on the use and implementation of molecular cytogenetic and NGS technologies. GEP: gene expression profile, LOH: loss of heterozygosity, CNAs: copy number alterations, iAMP: intrachromosomal amplification, SNP: single nucleotide polymorphism, GWAS: genome-wide association studies, WGS: whole-genome sequencing, WHO: World Health Organization, MTX: methotrexate, MRD: minimal residual disease, tNGS: targeted NGS.

**Figure 2 cancers-16-03965-f002:**
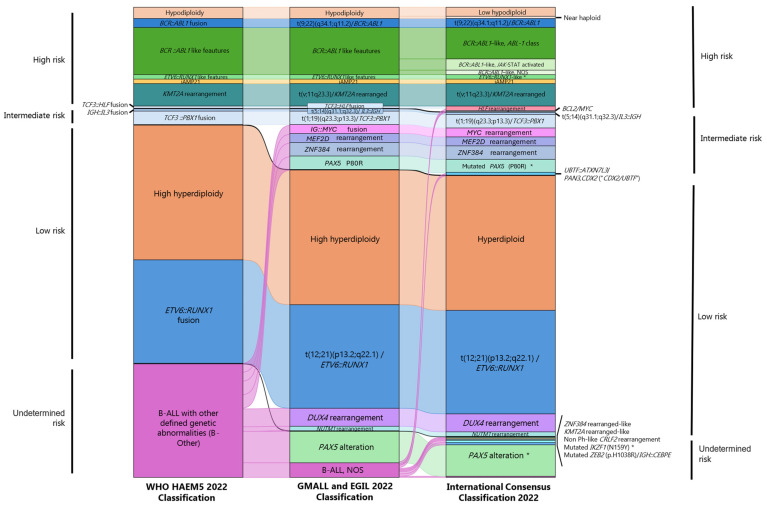
Recent ALL classification and risk subtypes based on genomic/transcriptomic profiles in child patients. As can be seen, genomic classification not only contributes to a better diagnosis but also has prognostic implications. Considerations: The width of the band corresponds to the percentage of cases with each of the alterations. An * indicates the provisional entity status according to the International Consensus Classification (2022).

**Figure 3 cancers-16-03965-f003:**
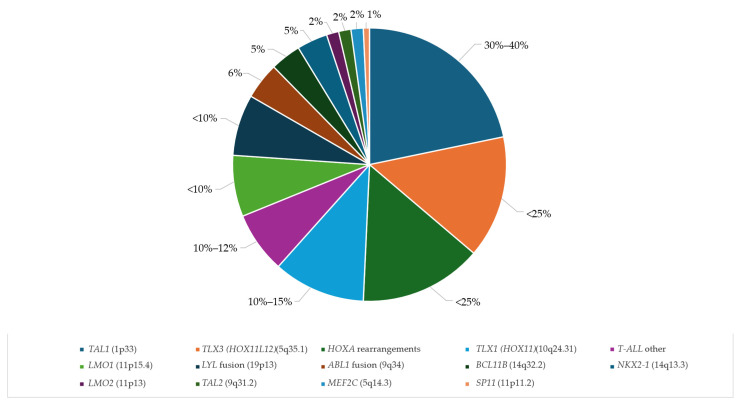
Frequency of recurrent alterations in T-ALL. Based on average values found in the reports analyzed in this review.

**Table 1 cancers-16-03965-t001:** Recent findings in ALL based on NGS applications.

Year	NGS	Sample Size	Application	Main Findings
2017	WES	1000 cases (B-ALL and T-ALL)	Therapeutic targets Identification	Clinical and molecular data at diagnosis and remission of ALL patients, including MRD are currently being collected. They will include the identification of somatic or germline variants associated with response to treatment, assessment of diversity, clonal evolution, etc. This study proposes to use NGS in diagnosis alongside morphological, immunophenotyping, and other molecular approaches, including scNGS [35].
2018	RNAseq	1223 B-ALL samples	Characterization of molecular subtypes for diagnostic accuracy and risk estimation	This study proposes fourteen molecular subtypes based on transcriptional profiling, including six previously undescribed subtypes: (i) *PAX5* or *CRLF2* fusions, (ii) *PAX5* mutation (p.P80R), (iii) *IKZF1* mutation (p.N159Y), (iv) *ZEB2* mutation (p.H1038R) or IGH/CEBPE fusion, (v) *HLF* rearrangements, and (vi) *NUTM1* rearrangements [36].
2019	RNAseq	170 B-ALL samples	Biomarker identification	Report on the first oncogenic subtype defined by a point mutation in ALL (*PAX5* P80R) that may harbor a second *PAX5* allele inactivation, biallelic *CDKN2A* deletion, and RAS signaling alterations [37].
2019	tNGS	17 childhood B-ALL samples	Translational research for diagnosis and monitoring of ALL patients	Matched diagnostic, remission, and relapse samples allowed for the detection of “potentially actionable gene variants” such as *CDKN2A/B* deletions and *KRAS*, *ABL1*, *FLT3,* and *NOTCH* mutations, as well as the detection of germline alterations such as *TP53* R248Q [38].
2019	RNA-seq	40 diagnoses and 42 relapsed BCP-ALL samples	Insights into epigenomic changes	At diagnosis, DUX4-specific lncRNAs were associated with the TGF-beta and p53 signaling pathways. For the Ph-like subtype, lncRNAs were correlated with genes involved in the JAK-STAT and PIK3/AKT/mTOR signaling pathways. For all subtypes, lncRNAs were found to be differentially expressed at diagnosis and relapse. About 80% of the lncRNAs were downregulated at relapse, e.g., *TCL6* in the DUX4r subtype, and, LINC00312 and miR-17-92a-1 in the DUX4r and Ph-like subtypes [39].
2020	WGS, ChIP-seq RNA-seq RRBS-Seq	34 B-ALL Samples	Epigenome in clonal evolution and treatment failure	This study provides the first reports of the role of epigenetic mechanisms in driving clonal evolution in B-ALL. Changes in gene expression were directly correlated with epigenetic alterations with changes in H3K27ac, and PRC2 (Polycomb complex) hypermethylation was detected at relapse in many patients [40].
2020	ATAC-Seq RNA-Seq	Primary cells of T-ALL patients and patient-derived xenograft models	Insights into epigenomic changes	Two distinct groups of T-ALL were identified based on accessible transcription factor (TF)-binding-motifs profiles. The first group was characterized by highly accessible *SPI1-*binding sites, and the second lacked hyper-accessible *SPI1*-binding sites and *DAB1* expression [41].
2021	RNAseq tNGS	58 AYAs and adult B-ALL samples	ALL diagnosis and stratification	This study introduces the first custom panel (LYmphoid Next-Generation Sequencing-LYNX Panel) as a versatile tool for the simultaneous analysis of genomic markers (mutations, indels, chromosomal aberrations, and IG/TR rearrangements) in most common lymphoid neoplasms (70 lymphoma-related genes) with a sensitivity of 5% variant allele frequency, as well as reliable identification of large genome-wide (≥6 Mb) and recurrent chromosomal aberrations (≥300 kb). This panel allowed for the description of novel variants classified as pathogenic or probable pathogenic in *TP53*, *NRAS*, *JAK2*, *PAX5*, *CREBBP*, *NF1*, *FLT3*, *ATM*, *KRA*S, and *RUNX1* genes, as well as IKZF1*^plus^* profile identification [42,43].
2021	in silico analysis	1978 samples of childhood B-ALL	Biomarker identification and prognostic association	*ADAM6* may be a novel genetic biomarker. Homozygous *ADAM6* deletion was detected around 60% of cases and was associated with higher total leukocyte count, higher first relapse rate, MRD-positive status, and higher mutation variants. It was more common in the *TCF3::PBX1* and *ETV6::RUNX1* subtypes [44].
2021	DNA-seq ddPCR	376 LPDs and 14 LPD cell lines	Technique Performance/Translational research	Assays were designed to capture the diversity and functional variation of joining genes across 1243 regions included in a tNGS panel. Sensitivity for detecting IG/TCR clonality ranged from 96.5 to 97.3%. The detection of CNAs was dependent on the locus, with high sensitivities being reported for trisomy 12 and del(11q) and lower sensitivities for del(17p) and del(13q) [45].
2021	DNA-seq RNA-seq	165 ALL patients at diagnosis	Technique performance/translational research	EuroClonality-NGS captures validation as a tNGS assay for SNVs, CNVs, Indels, Translocations, and IGH and TRC rearrangements. Less than half of the markers were also found in the RNAseq data (44%). The assay performance for somatic mutations was 100% reproducible for both sensitivity and specificity at >4% VAF [46].
2022	WGBS WGS WES	82 B-ALL patients	Insights into the epigenome	Local hypermethylation was observed in the global DNA methylation landscape of ALL and, more specifically, in T-ALL patients [47].
2022	Methyla-tionEPIC BeadChip Arrays	38 children with B-ALL pediatrics cases and controls	Methylation profile and microRNA changes landscape	Among specific genetic subtypes (*ETV6::RUNX1*, *TCF3::PBX1*, *IGH* rearrangement, hyperdiploidy), miRNAs with differentially methylated sites were found. The results showed that *MIR326*, *MIR200c*, *MIR125B*, *MIR203,* and *MIR181A* were significantly differentially expressed in B-ALL cases compared with healthy controls. This study was complemented by meta-analysis and in silico analysis to identify differentially expressed miRNAs in B-ALL. Although no consensus dataset was found among the studies, miR-181b, miR-181a, miR-128, miR-128a, miR-181c, miR-155, miR-142-3p, and miR-451 were postulated as biomarkers associated with B-ALL [48,49].
2023	ATAC-Seq ChIP-Seq Hi-C	154 B-ALL	Insights into epigenomic changes	This was the first large-scale analysis of chromatin accessibility in the B-ALL genome across extensive B-ALL subtypes. The percentage of differentially accessible chromatin sites (DAS) was associated with each molecular subtype. It was higher in the *KMT2A*r, *TCF3-PBX1* fusion, and *ZNF384*r subtypes. Analysis of bound TF motif footprint prevalence identified several ETS family TFs (EHF, ELF3, SPI1, and SPIB), zinc finger TFs (ZNF263, ZNF460, ZNF740, and ZNF148), and CTCF as the most altered motifs leading to differences in chromatin accessibility [50]. Subtypes: *BCR::ABL1*, *DUX4*r, *ETV6::RUNX1*, high hyperdiploidy, low hypodiploidy, *KMT2A*r, Ph-like, *PAX5*alt, *TCF3::PBX1*, *ZNF384*r, B-other samples, and B-ALL cell lines (697, BALL1, Nalm6, REH, RS411, SEM, and SUPB15)

TGS: targeted next-generation sequencing. scNGS: single-cell next-generation sequencing. WES: whole-exome sequencing. WGBS: whole-genome bisulfite sequencing. DAS: differentially accessible chromatin sites. MRD: minimal residual disease. scRRBS/RRBS-Seq: reduced-representation bisulfite sequencing or single-cell reduced-representation bisulfite sequencing. Hi-C: 3C-Seq to map chromatin contacts genome-wide or chromosome conformation capture. ddPCR: droplet digital PCR. DNAamp IG/TR assays developed by EuroClonality-NGS (DNA-amplicon-based methods). HeH: high hyperdiploidy. Potentially actionable genes include gene alterations that could lead to a newer molecular-targeted therapy, a change in the selection of drugs, or a change in patient or family counseling and management by identifying germline cancer-predisposing gene alterations. LPDs: lymphoproliferative disorders.

**Table 2 cancers-16-03965-t002:** Clinical utility of the implementation of NGS for ALL patients.

Method	Variants Detected	Genetic Subtypes	Limitations
WGS	Point mutations, aneuploidy, SV, CNVs, and BCR/TCR rearrangements. Useful for detecting germinal variants and polymorphism related to predisposition and treatment response, as well as GWAS	B-ALL: HeH, Hypodiploidy, *BCR::ABL1*, *ETV6::RUNX1; TCF3::PBX1*, *KMT2A*r, DUX4r; MEF2Dr; *ZNF384*r, *NUTM1*r; *HLF*r, *BCL2::MYC*; *PAX5* (P80R); *IKZF1* (N159Y); other related CNVs.	1. A considerable number of unknown significance or likely pathogenic status 2. Currently, this is considered a high-cost method based on sequencing-platform acquisition, data storage, and analysis.
		T-ALL: *BCL11B; TAL/LMO* rearrangements, *HOXA*; *SPI1; NKX2-1; TAL1* mutations	
WES	Point mutations, SV, CNVs, and aneuploidies. Useful for detecting novel fusions	B-ALL: *PAX5* (P80R), *IKZF1* (N159Y), and mutations in the Ph-like group (*ABL1*, *JAK*)	1. Detection of phenocopies 2. Limitation of the analysis to the detection of variants in coding regions 3. Less coverage (less sensitive)
Target Sequencing (DNA/RNA)	Point mutations, aneuploidy, SV (InDels and gene fusions), and CNVs. Useful for increasing sensitivity (greater depth of coverage compared with WGS studies) for the detection of variants with low VAF; as well as variants of intron- and splicing-related regions variants	Defined target variants, such as *BCR::ABL1*, *ETV6::RUNX1; TCF3::PBX1*, *BCL2::MYC*; intrachromosomal amplifications (iAMP21, *PAX5*^AMP^), and other variants, such as CNVs (*IKZF1*, *CDKN2A/2B*, *BTG1*, *PAX5*); or mutations including *PAX5* (P80R); or *IKZF1* (N159Y)	Non-targeted alterations
WTS	Point mutations, CNVs, gene fusion, GEP, alternative splicing analysis, and BCR/TCR rearrangements	Subtypes defined by distinct gene expression profiles (phenocopies). Target variants such as *BCR::ABL1*, *ETV6::RUNX1*, *TCF3::PBX1*, and *BCL2::MYC*, novel fusions, *PAX5* alteration and others	Ploidy alterations and NOS T-ALL or B-other ALL
		T-ALL: *SET::NUP214*, *PICALM::MLLT10*, *NUP98::RAP1GDS1*, *TAL/LMO* rearrangements	
OGM *	SV, aneuploidies, CNV, and balanced shifts in position	B-ALL: HeH, hypodiploidy, *BCR-ABL1*, *ETV6::RUNX1; TCF3::PBX1*, *KMT2A*r, DUX4r, *MEF2D*r; *ZNF384*r, *NUTM1*r, *HLF*r, *BCL2::MYC;* and other novel fusions	1. Point mutation detection 2. Only fresh or preserved samples that guarantee adequate uHMW DNA integrity can be used.
		T-ALL: *TAL/LMO* rearrangements *KMT2A::PRDM10* and other novel fusions	

Abbreviations. WGS: whole-genome sequencing, WES: whole-exome sequencing, WTS: whole-transcriptome sequencing, OGM: Optical Genome Mapping (GWAS: genome-wide association study, SV: structural variants, CNVs: copy number variants), HeH: high hyperdiploidy, BCR: B-cell receptor; TCR: T-cell receptor; uHMW: ultra-high molecular weight, NOS: not otherwise specified. OGM* is not an NGS technique; it does not determine the nucleotide sequence, but it allows for the analysis of the genome’s structure and organization.

**Table 3 cancers-16-03965-t003:** Comparison between different leukemia classifications based on genomic profiles.

WHO-HAEM5, 2022 Classification [55]	MLL, GMALL, and EGIL, 2022 Classification [57]	International Consensus Classification (ICC), 2022 [58]
**B-acute lymphoblastic leukemia (B-ALL)**		
**Aneuploidies**		
High hyperdiploidy	High hyperdiploidy	Hyperdiploidy
Hypodiploidy	Hypodiploidy	Low hypodiploidy
		Near haploid
**Gene fusion**		
*BCR::ABL1* fusion	t(9;22)(q34.1;q11.2)/*BCR::ABL1*	t(9;22)(q34.1;q11.2)/*BCR::ABL1*
		with lymphoid only involvement
		with multilineage involvement
*ETV6::RUNX1* fusion	t(12;21)(p13.2;q22.1)/*ETV6::RUNX1*	t(12;21)(p13.2;q22.1)/*ETV6::RUNX1*
*IGH::IL3* fusion	t(5;14)(q31.1;q32.3)/*IL3::IGH*	t(5;14)(q31.1;q32.3)/*IL3::IGH*
*TCF3::PBX1* fusion	t(1;19)(q23.3;p13.3)/*TCF3::PBX1*	t(1;19)(q23.3;p13.3)/*TCF3::PBX1*
*TCF3::HLF* fusion	*TCF3::HLF* fusion	
		*BCL2/MYC*
**Phenocopies**		
*BCR::ABL1*-like features	*BCR::ABL1–*like	*BCR::ABL1*–like, *ABL-1* class rearranged
		*BCR::ABL1–*like, JAK-STAT activated
		*BCR::ABL1*–like, NOS
*ETV6::RUNX1*-like features	*ETV6::RUNX1-*like	*ETV6::RUNX1*-like (provisional entity)
		*ZNF384* rearranged-like (provisional entity)
		*KMT2A* rearranged-like (provisional entity)
**Intrachromosomal amplifications**		
iAMP21	iAMP21	iAMP21
**Other specified rearrangements**		
*KMT2A* rearrangement	t(v;11q23.3)/*KMT2A* rearranged	t(v;11q23.3)/*KMT2A* rearranged
	*IG::MYC* fusions	*MYC* rearrangement
B-Other	*DUX4* rearrangement	*DUX4* rearrangement
	*MEF2D* rearrangement	*MEF2D* rearrangement
	*ZNF384(362)* rearrangement	*ZNF384(362)* rearrangement
	*NUTM1* rearrangement	*NUTM1* rearrangement
		*HLF* rearrangement
		UBTF::ATXN7L3/PAN3,CDX2 (“CDX2/UBTF”)
		Non Ph-like *CRLF2* rearrangement
**Point mutations**		
	*PAX5* P80R	Mutated *PAX5* (P80R) (provisional entity)
		Mutated *IKZF1* (N159Y)
		Mutated *ZEB2* (p.H1038R)/*IGH::CEBPE* (provisional entity)
**Other B-ALL subtypes**		
	*PAX5* alteration	*PAX5* alteration (provisional entity)
	B-ALL, NOS	B-ALL, NOS
**T-acute lymphoblastic leukemia (T-ALL)**		
Early T-cell precursor ALL, NOS	Early T-cell precursor ALL, NOS	Early T-cell precursor ALL, BCL11B-activated
T-ALL, NOS	T-ALL, NOS	Early T-cell precursor ALL, NOS
		T-ALL, NOS

WHO-HAEM5, 2022 Classification. World Health Organization. B-lymphoblastic leukemia/lymphoma classification. MLL. Münchner Leukämielabor. GMALL. German Multicenter Study group for Adult Acute lymphoblastic leukemia. EGIL. European Group for Immunological Classification. NOS: not otherwise specified (when diagnostic criteria based on cytogenetic and molecular testing are applied, but subtyping is not possible even with exhaustive testing).

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
