# Peer review of "Unraveling the Genetic Heterogeneity of Acute Lymphoblastic Leukemia Based on NGS Applications"

_cancers, 2024, doi:10.3390/cancers16233965_

Round 1
Reviewer 1 Report
Comments and Suggestions for Authors
This review article covers important genetic heterogeneity of acute lymphoblastic leukemia (ALL) based on NGS applications.
This specific strategy presented in this review article is designed to describe and recommend recent advances in NGS and their implication into clinical practice. All new subtypes of (ALL) and their prognostic values and genomic information related to oncogenic processes capable to establish and propose new therapies.
The compiled data are supported with 3 very informative tables and 3 important figures. The review article concludes with 165 very recent literature references. This consolidated study constitutes crucially important developments, which were never ever reported and in such systematic and specific order and sequences, therefore deserve publication as soon as possible.
The following suggested changes and recommendations should be introduced before the publication of the manuscript:
• Page 4, Figure 1. It is highly suggested that authors should move the figure 1 on page 3, line 100, where the description of (NGS) is discussed. That would enhance the value of the figure and explore an additional sequential scoring as conclusion from the presented representative images.
• Page 18, figure 3, move to page 17 Line 566. That would increase flow of information and solidify the important outcome of good correlation and recurrent alerations in T-ALL.
• Page 19, Line 604. Conclusions. This section must be expanded, as it is too short for this type of review. In its present format, the authors do not fully describe the desired/anticipated effects of (ALL) and recent advances in NGS. Authors should also include comparative data available in the literature regarding existing clinical developments of NGS technologies demonstrating the high prognostic value of (ALL).
The manuscript is of good quality, well-written, and meets the standard for articles published in Cancers. I recommend it for publication after the correction of these minor suggested changes.
Author Response
Reviewer #1: This review article covers important genetic heterogeneity of acute lymphoblastic leukemia (ALL) based on NGS applications.
This specific strategy presented in this review article is designed to describe and recommend recent advances in NGS and their implication into clinical practice. All new subtypes of (ALL) and their prognostic values and genomic information related to oncogenic processes capable to establish and propose new therapies.
The compiled data is supported with 3 very informative tables and 3 important figures. The review article concludes with 165 very recent literature references. This consolidated study constitutes crucially important developments, which were never ever reported and in such systematic and specific order and sequences, therefore deserve publication as soon as possible.
We appreciate that the reviewer found merit in our work, and we are grateful for the helpful comments, which we have addressed in the revised version of the manuscript. In our view, these comments have enabled us to improve the manuscript significantly (please find our responses to the reviewer’s comments below).
1 Changes made following Reviewer 1's suggestions:
- Figure 1 has been moved to line 105 (page 4) in the section NGS Applications in ALL.
- Figure 3 has been relocated to page 19, line 591, which enhances the flow of information on recurrent alterations in T-ALL.
- In the conclusion, we emphasize that some NGS applications are now being adopted in clinical practice and have already been included in recommendations and guidelines such as NCCN (2024) and ELN (2022), as well as in protocols and clinical trials where research findings are used in decision-making for managing patients with ALL. The practical use of these technologies is also discussed in the introduction, as well as in the description of each subtype where prognosis is highlighted.
Reviewer 2 Report
Comments and Suggestions for Authors
The paper has no major flaws but there are many similar reveiew papers, some from the gruops that contributed to the discovery, so there is no gap in the literatute to be filled in by the manuscript. I leave this problem to the editor' discretion.
Some issues with teh text:
It should be better explained /presented which are routine methods implemented in current treatment protocols and which are experimental or developmental, such as OGM, Ataq-seq. This is mixed by the authors and make the paper less useful in everyday practice.
It should be stressed /noted that so far NGS-based methods are variable capacity to dect CNV and rearrangements. In gerenarl thay should be used in conjunction with karytyping and CNV analysis.
What is the current practical status of NGS methods in ALL -genotyping and prognostication? What pathaway is edovocated by the authors? What combination of studies is curret gold standard?
What is the practice in their centre?
Author Response
Reviewer #2:
- “It should be better explained/presented which are routine methods implemented in current treatment protocols and which are experimental or developmental, such as OGM, ATAC-seq. This is mixed up by the authors and makes the paper less useful in daily practice.”
The integration of these techniques into clinical practice remains limited. However, as OGM is an emerging technique with potential clinical applications demonstrated over the past three years, we have included this information in Table 2, and on page 9, line 153 to 166. This includes recommendations from the International Consortium for OGM in Hematologic Malignancies and the latest publication on OGM validation and guidance for clinical implementation. Two new references have been added (Ref 51,52, line 163).
[51] Levy B, Kanagal-Shamanna R, Sahajpal NS, Neveling K, Rack K, Dewaele B, Olde Weghuis D, Stevens-Kroef M, Puiggros A, Mallo M, Clifford B, Mantere T, Hoischen A, Espinet B, Kolhe R, Solé F, Raca G, Smith AC. A framework for the clinical implementation of optical genome mapping in hematologic malignancies. Am J Hematol. 2024 Apr;99(4):642-661. doi: 10.1002/ajh.27175. Epub 2024 Jan 2. PMID: 38164980.
[52] Sahajpal NS, Mondal AK, Tvrdik T, Hauenstein J, Shi H, Deeb KK, Saxe D, Hastie AR, Chaubey A, Savage NM, Kota V, Kolhe R. Clinical Validation and Diagnostic Utility of Optical Genome Mapping for Enhanced Cytogenomic Analysis of Hematological Neoplasms. J Mol Diagn. 2022 Dec;24(12):1279-1291. doi: 10.1016/j.jmoldx.2022.09.009. Epub 2022 Oct 17. PMID: 36265723.
- “It should be noted that so far NGS-based methods have a variable ability to detect CNVs and rearrangements. In general, they should be used in conjunction with karyotyping and CNV analysis.”
We agree with the reviewer’s comments. NGS-based methods indeed have a variable ability to detect CNVs and rearrangements. Currently, as analysis tools continue to evolve, CNV analysis should be used in conjunction with karyotyping and other CNV technologies, such as SNP arrays, MLPA, or OGM. Additionally, data analysis today should incorporate comparisons across various algorithms, especially when using targeted panels. This consideration takes into account the different bioinformatics methods that can enhance CNV identification. The information related to this reviewer’s suggestion is highlighted in lines 167 to 170. Two recent references have been added to support this information (references 53 and 54, line 168).
[53] Gordeeva V, Sharova E, Arapidi G. Progress in Methods for Copy Number Variation Profiling. Int J Mol Sci. 2022 Feb 15;23(4):2143. doi: 10.3390/ijms23042143. PMID: 35216262; PMCID: PMC8879278.
[54] Song Y, Fang Q, Mi Y. Prognostic significance of copy number variation in B-cell acute lymphoblastic leukemia. Front Oncol. 2022 Aug 4;12:981036. doi: 10.3389/fonc.2022.981036. PMID: 35992882; PMCID: PMC9386345.
3.What is the current practical situation of NGS methods in ALL - genotyping and prognosis? What pathway do the authors advocate? What combination of studies is the current gold standard?
Currently, NGS methods in ALL (acute lymphoblastic leukemia) play an increasingly important role in both genotyping and prognostic evaluation. While NGS effectively detects a broad range of genetic alterations, it remains complementary to traditional cytogenetic methods due to limitations in identifying certain structural variants and copy number variations (CNVs). The authors advocate a combined approach, where NGS is used alongside established cytogenetic techniques, such as karyotyping, fluorescence in situ hybridization (FISH), and SNP arrays, to enhance diagnostic precision and prognostic assessment. This combination approach is considered the current gold standard in ALL genotyping, offering a comprehensive understanding of the genomic landscape essential for accurate risk stratification and personalized treatment planning.
Recommendations for the clinical application of NGS and complementary methods are based on guidelines such as the NCCN (2024) and the European LeukemiaNet (ELN) recommendations (references 22 and 29). Additionally, the document “Diagnosis, prognostic factors, and assessment of ALL in adults: 2024 ELN recommendations from a European expert panel” provides an overview of the recommended studies for diagnosis, prognosis refinement, and patient management. This reference had already been included in the introduction (line 90 to 92)
Regarding the current practical situation of NGS methods in ALL - genotyping and prognosis, the conclusion includes the main principal uses of NGS in ALL.
- What is the practice in their center?
Dr. Hernández Rivas leads the Oncology Cytogenetics Unit (UCO) in the Hematology Service at the University Hospital of Salamanca. The unit integrates studies of molecular cytogenetics, Next Generation Sequencing (NGS), and Optical Genome Mapping (OGM). Our center is part of the Spanish PETHEMA network.
PETHEMA (Spanish Hematology and Hemotherapy Research Group) is a network dedicated to advancing the diagnosis and treatment of Acute Lymphoblastic Leukemia (ALL) in Spain. It brings together a multidisciplinary team of experts from over 100 hospitals nationwide, focusing on clinical, molecular, and genetic research. The network aims to optimize patient care through collaborative clinical trials, the development of personalized therapies, and the integration of advanced diagnostic techniques such as molecular cytogenetics, NGS, and OGM. PETHEMA also works to standardize practices, improve prognosis, and ensure the best treatment outcomes for ALL patients across the country.
Currently, we are participating in the PETHEMA LAL19 trial (NCT 04179929), which aims to establish the genetic subtype (primary alteration) of adult BCP ALL patients enrolled in the trial and correlate these with measurable residual disease (MRD) levels and survival. Specifically, the LAL19 trial includes Ph-negative patients aged 18-65 years: those with MRD ≥0.01% at day +35 or high-risk genetics receive alloHSCT, while MRD <0.01% patients with standard-risk genetics receive maintenance chemotherapy. Genetic analyses are centralized: FISH and NGS DNA panels are performed at the University Hospital of Salamanca, RNAseq panels at the Hospital 12 de Octubre, FISH panels at Hospital La Fe, and SNP arrays at the Josep Carreras Institute/ICO-Hospital Germans Trias i Pujol. MRD determinations are centrally conducted using next-generation flow cytometry at the Cytometry Service, NUCLEUS, University of Salamanca.
In order to provide an appropriate response to the three questions from Reviewer 2, the conclusions section has been developed and expanded on page 20.
Reviewer 3 Report
Comments and Suggestions for Authors
The manuscript is a comprehensive and very thorough analysis on the novel contributions of the cutting edge NGS applications to understanding the genetic heterogeneity in Acute Lymphoblastic Leukemia. The manuscript is well organized and presented clearly and the new sets of data are integrated with current clinical and molecular knowledge. The authors produced a very important and interesting manuscript that will be of high value to the scientific community.
Some minor details:
- Line 191 – “BRC::ABL1” should be “BCR::ABL1”
- Line 202 – “Ph+ LAL” should be “Ph+ALL”
Author Response
Comments: The manuscript is a comprehensive and very thorough analysis on the novel contributions of the cutting edge NGS applications to understanding the genetic heterogeneity in Acute Lymphoblastic Leukemia. The manuscript is well organized and presented clearly and the new sets of data are integrated with current clinical and molecular knowledge. The authors produced a very important and interesting manuscript that will be of high value to the scientific community.
Some minor details:
-Line 191 – “BRC::ABL1” should be “BCR::ABL1”
-Line 202 – “Ph+ LAL” should be “Ph+ALL”
Response: We would like to thank the reviewer for the encouraging comments, and we have implemented all the suggestions and made the suggested changes in the manuscript, writing BCR::ABL1 and Ph+ALL.
Reviewer 4 Report
Comments and Suggestions for Authors
1. What is the relevance and the association of KIT, CSF3R, TNFα, CSF2RA, CSF1R, MS4A1/CD20, and IL7R genes in ALL with clinical characterstic of the patients diagnosed with ALL.
2. Please explain the mechanism of PAX5, TP53, IKZF1, CDKN2A/2B or 79 RB1 in B-ALL, and NOTCH1, FBXW7, or WT1 in T-ALL.
Comments on the Quality of English LanguageIt seems fine
Author Response
- What is the relevance and the association of KIT, CSF3R, TNFα, CSF2RA, CSF1R, MS4A1/CD20, and IL7R genes in ALL with clinical characteristics of the patients diagnosed with ALL?
The information on the overexpression of these genes was included in the text because it is considered significant, as it allowed a group of researchers to demonstrate that BCR::ABL1-positive patients may be more diverse than previously known (references 61 and 62). Furthermore, these variations could have prognostic value related to treatment response (reference 57). While these details are not described in depth due to character limitations, we briefly highlight some relevant facts that justify mentioning these findings in our review:
KIT gene is a type-III receptor tyrosine kinase that contributes to signal transduction in hematopoietic stem cells and plays an important role in self-renewal and differentiation into myeloid and lymphoid cells. Some studies show that gain-of-function mutations cause systemic mastocytosis. Although prognostic effects have been identified in other acute leukemias, such as AML, no recent studies have reported an association between the overexpression of this gene and any clinical features in acute lymphoblastic leukemia. However, its functional role may still be explored in specific subtypes such as BRC:ABL1 ALL.
- Weidemann RR, Behrendt R, Schoedel KB, Müller W, Roers A, Gerbaulet A. Constitutive Kit activity triggers B-cell acute lymphoblastic leukemia-like disease in mice. Exp Hematol. 2017 Jan;45:45-55.e6. doi: 10.1016/j.exphem.2016.09.005. Epub 2016 Sep 21. PMID: 27664314
- Katagiri S, Chi S, Minami Y, Fukushima K, Shibayama H, et al. Mutated KIT Tyrosine Kinase as a Novel Molecular Target in Acute Myeloid Leukemia. J. Mol. Sci. 2022 April 23; 23(9), 4694; https://doi.org/10.3390/ijms23094694
Regarding the CSF3R gene, mutations such as S661N, S557G, and Q170X may be associated with disease progression. However, studies on the effect and prognostic value of these variants have limitations in terms of sample size, and should be compared with other subgroups, even though they have already been evaluated in 9p- patients.
- Rashid M, Alasiri A, Al Balwi MA, Alkhaldi A, Alsuhaibani A, Alsultan A, Alharbi T, Alomair L, Almuzzaini B. Identification of CSF3R Mutations in B-Lineage Acute Lymphoblastic Leukemia Using Comprehensive Cancer Panel and Next-Generation Sequencing. Genes (Basel). 2021 Aug 27;12(9):1326. doi: 10.3390/genes12091326. PMID: 34573308; PMCID: PMC8470887.
TNF-α is involved in all steps of leukemogenesis, including cell transformation, proliferation, angiogenesis and extramedullary infiltration, as well as in immune evasion, survival and treatment resistance of malignant cells. It may be related to progression and relapse in acute leukemia and some polymorphisms have even been reported to be associated with a worse prognosis.
- Verma S, Singh A, Yadav G, Kushwaha R, Ali W, Verma SP, Singh US. Serum Tumor Necrosis Factor-Alpha Levels in Acute Leukemia and Its Prognostic Significance. Cureus. 2022 May 8;14(5):e24835. doi: 10.7759/cureus.24835. PMID: 35547942; PMCID: PMC9090230
- El Baiomy E MA, Tamer A, El Menshawy N, El-Sebaie AH, et al. The prognostic values of the IL-10 (G1082A) and TNF-α (G308A)polymorphisms in Egyptian patients with acute lymphoblastic leukemia: A single-center study. Indian Jour. Can. 2023 April; 60(2):p 217-223. doi: 10.4103/ijc.IJC_102_21
CSF1R signaling is necessary for the survival, proliferation, and differentiation of many myeloid cell types. In acute lymphoblastic leukemia it has been found to be altered in high-risk subgroups such as Ph-like. The SSBP2-CSF1R fusion is considered a potential biomarker in this subgroup and may be associated with poor prognosis and treatment resistance. In addition, studies in animal models have shown that it may play a key role in the regulatory axis between leukemia cells and macrophages and delays leukemia progression.
- Wang H, Wang Y, Hao L, Liu X, Zhang J, Yao P, Liu D, Wang R. Treatment for a primary multidrug-resistant B-cell acute lymphoblastic leukemia patient carrying a SSBP2-CSF1R fusion gene: a case report. Front Oncol. 2023 Nov 30;13:1291570. doi: 10.3389/fonc.2023.1291570. PMID: 38107066; PMCID: PMC10723836
- Tasian SK, Loh ML, Hunger SP. Philadelphia chromosome–like acute lymphoblastic leukemia. Clinical Trials & Observations. Check for Updates. Blood. 2017;30 (19): 2064–2072. https://doi.org/10.1182/blood-2017-06-743252
- Li K, Xu W, Lu K, Wen Y, Xin T, Shen Y, Lv X, Hu S, Jin R, Wu X. CSF-1R inhibition disrupts the dialog between leukaemia cells and macrophages and delays leukaemia progression. J Cell Mol Med. 2020 Nov;24(22):13115-13128. doi: 10.1111/jcmm.15916. Epub 2020 Oct 10. PMID: 33037771; PMCID: PMC7701573.
MS4A1/CD20 is a B cell differentiation antigen with variable expression in B cell precursor acute lymphoblastic leukemia. Its expression has generally been associated with an adverse prognosis, but results need to be validated in cohorts with harmonized treatment protocols, and evaluated in both pediatric and adult patient groups.
- Alduailej H, Kanfar S, Bakhit K, Raslan H, Alsaber A, et al. Outcome of CD20-positive Adult B-cell Acute Lymphoblastic Leukemia and the Impact of Rituximab Therapy. Clinicl Lymp Myelo and Leukemia. 2020 sep; 20(9):e560-e568
- Serbanica AN, Popa DC, Caruntu C, Pasca S, Scheau C, et al. The Significance of CD20 Intensity Variance in Pediatric Patients with B-Cell Precursor Acute Lymphoblastic Leukemia. Clin. Med. Feb 2023, 12(4), 1451; https://doi.org/10.3390/jcm12041451
IL-7R expression is a functional biomarker of T-ALL cells with leukemia-initiating potential and plays a crucial role in T-ALL pathogenesis. Mutations promote cell transformation and tumor formation. Overall, the findings indicate that IL7R mutational activation is involved in human T-cell leukemogenesis, it remains unclear and further studies should be performed in high-risk subgroups such as BCR::ABL1.
- Zenatti PP, Ribeiro D, Li W, Zuurbier L, Silva MC, Paganin M, Tritapoe J, Hixon JA, Silveira AB, Cardoso BA, Sarmento LM, Correia N, Toribio ML, Kobarg J, Horstmann M, Pieters R, Brandalise SR, Ferrando AA, Meijerink JP, Durum SK, Yunes JA, Barata JT. Oncogenic IL7R gain-of-function mutations in childhood T-cell acute lymphoblastic leukemia. Nat Genet. 2011 Sep 4;43(10):932-9. doi: 10.1038/ng.924. PMID: 21892159; PMCID: PMC7424552.
- Almeida, A.R.M., Neto, J.L., Cachucho, A. et al. Interleukin-7 receptor α mutational activation can initiate precursor B-cell acute lymphoblastic leukemia. Nat Commun 12, 7268 (2021). https://doi.org/10.1038/s41467-021-27197-5
- Please explain the mechanism of PAX5, TP53, IKZF1, CDKN2A/2B or RB1 in B-ALL, and NOTCH1, FBXW7, or WT1 in T-ALL.
The roles of genes were not exhaustively detailed in this review, as they fall outside its primary focus of the review, and due to character limitations in the document. However, we have provided a concise description of several relevant genes below, which we would be happy to include in the main text if the editor finds it appropriate.
PAX5 is a master regulator of B cell differentiation and maturation process, and its biological functions, as well as its role in the leukemogenesis, have been widely studied. However, the effects of its variants need further clarification, as they depend on the type of alteration (e.g., haploinsufficiency or gain-of-function). The ICC 2022 has proposed subgroups based on gene expression profiles such as P80R or PAX5alt. However, even within this latter group, the biological effect should be analyzed taking into consideration the type of mutation (SNVs, CNVs, fusions,) and other concomitant variants, such as RAS pathway gene mutations, tumor suppressors mutations, and others.
- Jia Z, Gu Z. PAX5 alterations in B-cell acute lymphoblastic leukemia. Front Oncol. 2022 Oct 25;12:1023606. doi: 10.3389/fonc.2022.1023606. PMID: 36387144; PMCID: PMC9640836.
- Gruenbacher S, Jaritz M, Hill L, Schäfer M, Busslinger M. Essential role of the Pax5 C-terminal domain in controlling B cell commitment and development. J Exp Med. 2023 Dec 4;220(12):e20230260. doi: 10.1084/jem.20230260. Epub 2023 Sep 19. PMID: 37725138; PMCID: PMC10509461
IKZF1 is a transcription factor that acts as a critical regulator of lymphoid differentiation participating in nucleosome remodeling and the deacetylase complex, transcription initiation, and others processes. It is frequently altered in pediatric and adult patients, with deletion being its most common alteration. In ALL, IKZF1 alterations are associated with a lower response to treatment (TKI and glucocorticoids), and poor prognosis.
- Marke R, van Leeuwen FN, Scheijen B. The many faces of IKZF1 in B-cell precursor acute lymphoblastic leukemia. Haematologica. 2018 Apr;103(4):565-574.doi: 10.3324/haematol.2017.185603. Epub 2018 Mar 8. PMID: 29519871; PMCID: PMC5865415.
- Vairy S, Tran TH, IKZF1 alterations in acute lymphoblastic leukemia: The good, the bad and the ugly. Blood Reviews. 202. 44:100677. doi:/10.1016/j.blre.2020.100677.
Although variants in CDKN2A, TP53 and RB1 are frequent in some ALL subtypes, their alterations are frequent in other types of cancer and their value as prognostic biomarkers is still discussed. In most cases of ALL, CDKN2A is inactivated by homozygous deletions, or appears concomitantly with deletions of IKZF1 or PAX5. It has been hypothesized that these deletions occur due to partial monosomies of chromosome 9, or hypermethylation of the gene's promoter region. TP53 mutations are associated with poorer overall survival (OS) and event-free survival (EFS) in both pediatric and adult patients, and are more frequent in subgroups with low hypodiploidy. RB1 deletions and mutations also are more frequent in this subgroup.
Similarly, NOTCH1 activation plays a critical role in multiple stages of T cell development, analogous to the regulatory role of PAX5 in B cells. NOTCH1 gain-of-function mutations are highly frequent in T-ALL (present in around 60% of cases) and, similar to mutations in FBXW7, are often associated with poorer treatment response (Notch pathway alterations). However, some studies suggest that NOTCH1 mutation may predict a favorable outcome in pediatric T-ALL. Therefore, its prognostic value should continue to be investigated and compared across studies.
Round 2
Reviewer 2 Report
Comments and Suggestions for Authors
I acknowledge the explanations of the authors. Also the manuscript is now improved. It is now ready for publication.